



# Retrieving horizontally resolved wind fields using multi-static meteor radar observations

Gunter Stober[1], Jorge L. Chau[1], Juha Vierinen[2], Christoph Jacobi[3], and Sven Wilhelm[1]

[1]Leibniz-Institute of Atmospheric Physics, Schlossstr. 6, 18225 Kühlungsborn, Germany
[2]The Artic University of Norway, Department of Physics and Technology, Norway
[3]University of Leipzig, Institute for Meteorology, Stephanstr. 3, 04103 Leipzig, Germany

*Correspondence to:* stober@iap-kborn.de

**Abstract.** Recently, the MMARIA (Multi-static, multi-frequency Agile Radar for Investigations of the Atmosphere) concept of a multi-static VHF meteor radar network to derive horizontally resolved wind fields in the mesosphere/lower thermosphere was introduced. Here we present preliminary results of the MMARIA network above Eastern Germany using two transmitters located at Juliusruh and Collm, and 5 receiving links two monostatic and three multi-static). The observations are complemented during a one-week campaign, with a couple of addition continuous-wave coded transmitters, making a total of 7 multi-static links. In order to access the kinematic properties of non-homogenous wind fields we developed a wind retrieval algorithm that applies regularization to determine the non-linear wind field in the altitude range of 82-98 km. The derived horizontally resolved wind fields are compared to wind fields retrieved by a more established volume velocity processing that includes the horizontal gradients of the horizontal wind components. The potential of such observations and the new retrieval to investigate gravity waves with horizontal scales between 50-200 km is presented and discussed.

## 1 Introduction

The upper mesosphere/lower thermosphere (MLT) is a highly dynamic region dominated by a variety of waves (gravity waves, tides, planetary waves) covering different spatial and temporal scales. To characterize this variability, it is desirable to develop remote sensing techniques to retrieve horizontally resolved structures from continuous observations. A particular challenge is the determination of horizontally resolved wind fields at mesospheric altitudes, needed to access small scale variations associated to gravity waves (GW). GWs are considered to be a major driver of MLT dynamics as they carry energy and momentum from other (mainly lower) atmospheric layers into the mesosphere (Fritts and Alexander, 2003; Becker, 2012).

Over the past decades specular meteor radars (SMRs) have become a reliable and wide spread tool to investigate mesospheric mean winds (e.g., Elford, 1959; Roper, 1975; Nakamura et al., 1991; Hocking et al., 2001; Hall et al., 2005; Jacobi et al., 2009; Stober et al., 2017; McCormack et al., 2017; Wilhelm et al., 2017). These systems are also capable of provide valuable information about gravity waves and tides (Fritts et al., 2010a; Jacobi, 2011) as well as to estimate the gravity wave momentum flux (e.g., Hocking, 2005; Fritts et al., 2010b; de Wit et al., 2014; Placke et al., 2015). In particular, Fritts et al. (2012) pointed out that the quality of the gravity wave momentum flux strongly depends on the number of meteor detections per time interval and the diameter of the observation volume.



The spatial and temporal intermittency of GWs are hardly accessible from point measurements. Airglow imagers (e.g., Hecht et al., 2000, 2007) or the Advanced Mesospheric Temperature Mapper (Taylor et al., 2009; Pautet et al., 2014) are able to observe small scale GWs as intensity or temperature fluctuations. However, due to the often missing background wind information the intrinsic GW properties can not be inferred. Smith et al. (2017) combined the 2D airglow information with MR

wind measurements to investigate a bore event across Europe and derived the intrinsic properties of the GW. At present there are only a few attempts to measure horizontally resolved wind structures on comparable scales to airglow imagers by radars (Stober et al., 2013).

Recently, Stober and Chau (2015) introduced the MMARIA (Multi-static Multi-frequency agile radar for investigation of the atmosphere) concept to observe horizontally resolved wind fields combining monostatic and mutli-static SMR observations.

The MMARIA concept allows an increase number of detected meteors per transmitter, an extended altitude coverage, and more even spatial sampling within the field of view, when compared to traditional monostatic SMRs. They demonstrated the potential to access the kinematic properties of non-homogenous wind fields applying volume velocity processing (VVP) (Waldteufel and Corbin, 1979) to the multi-static SMR observations, i.e., by deriving the horizontal gradients of the horizontal wind components, in addition to their mean values. The multi-static observation geometry allows the observation of almost the

same measurement volumes from different angles. Thus, it is possible to access the first order inhomogeneities of the mesoscale wind field, e.g., horizontal divergence, relative vorticity, stretching and shearing deformation. Here we a re going to extent the existing approach to the retrieval of arbitrary wind fields using multi-static observations of meteor radar networks. Figure 1 presents a schematic of such an network.

The paper is structured as follows. In section 2 we present a summary of the normal meteor radar wind retrieval. This

method is going to be expanded in section 3 to horizontally resolved winds in a full Earth geometry. In section 4 we perform an initial validation and consistency check. The potential use of the new horizontally resolved wind retrieval is given in section 5 presenting first horizontal wavelength spectra. Our main conclusions are presented in the last section. The appendix contains all equations required for the WGS84 coordinate transformations.

## 2   Wind analysis of mean meteor radar winds

Meteors entering the Earth's atmosphere form an ambipolar plasma trail, if they are fast and heavy enough. The trail is drifted by the ambient neutral winds at the altitude of its deposition. Combining the radial Doppler measurement with radar interferometry (Jones et al., 1998) permits the measurement of the radial velocity, range, and angle of arrival. Due to the huge number of meteor detections in the course of a day it is possible to determine the prevailing wind speeds by binning the observed meteors in space (vertical) and time applying a so called all-sky fit (Hocking et al., 2001; Holdsworth et al., 2004) over typically one hour in

time and a few kilometers in height.

To increase the robustness of the standard wind fit estimation, e.g., better temporal resolution and altitude coverage, it is possible to use regularization by adding constraints based on a priori information. Recently, we have implemented a routine to derive mesospheric winds using full non-linear error propagation, an additional weighting to account for sampling effects,





and a smoothness regularization. The error propagation is straightforward as most of the available systems provide errors for the radial velocity measurements. The angular uncertainties are estimated to be in the order of $2°$ (Jones et al., 1998). Further, the errors of the 3D wind components are estimated from the covariance matrix and updated with each iteration. The sampling effects are mainly caused by meteors occurring randomly in space and time, but we use a fixed grid in time and altitude.

Therefore, we apply an additional weighting, with a Gaussian kernel, to account for altitude and time differences between actual occurrence of a meteor and the reference time and altitude. The half width of the Gaussian kernel is given by the width of the altitude and time bins.

The smoothness regularization scheme consists of the vertical and temporal derivative for each wind component taken as constant within each bin. The local derivatives are computed from the gridded data, viz. the temporal derivative is estimated

using the time bins before and after, the vertical derivative is defined by using one altitude bin below and above. The initial guess is given by a standard least squares solution without any regularization constraint. This solution is then used for the next iteration and the new solution is constrained by the previous one. Typically we need 3-5 iterations to achieve convergence. Basically our wind estimates do not change more than the statistical uncertainty, which is in the order of 1-6 m/s at altitudes between 82-95 km. At the edges of the meteor layer (below 82 km or above 95 km) the error can reach up to 15-20 m/s as the

number of meteors used for each wind estimate drops significantly at these altitudes. More details about the application of this algorithm can be found in Stober et al. (2017).

## 3  Wind analysis of arbitrary wind fields

### 3.1  Implementing the Earth geometry in the wind analysis

A new aspect of the MMARIA concept is the necessity to consider the geometry of the Earth. At present our domain area in

Germany has a horizontal extension of approximately $600 \times 600$ km. These distances are too large to assume a plane geometry, which is very often the case for classical monostatic MRs, where all observations are referred to the location of the radar itself. However, it is straight-forward to at least consider that the altitude or height above the surface needs to be corrected for the Earth curvature using a mean Earth radius ($R_E = 6378137.0$ m). Further, it is possible to obtain a local elevation angle for each meteor, instead of the one observed relative to the receiver location. However, this simple corrections turned out to still

be insufficient dealing with large domain areas. Therefore, we outline a more detailed procedure taking into account the Earth shape with the WGS84 geoid model (National Imagery and Mapping Agency, 2000).

In the following we outline the procedure how to compute new local coordinates (ENU: East-North-Up) for each meteor to reduce potential errors in the wind field estimation due to projection issues. Considering that our Earth is not a perfect sphere we have to deal with two different coordinates, the geodetic coordinates (longitude and latitude) and the Earth-Centered,

Earth-Fixed (ECEF) coordinates, also called geocentric coordinates. The geodetic coordinates are determined by the normal to the ellipsoid, whereas the ECEF coordinates are defined by the Earth center using a (X, Y, Z)- coordinate system. Thus, the geodetic and geocentric latitude can be different.



We need to transform the observed meteor positions relative to the radar into a local coordinate frame (ENU) by determining the geodetic longitude, latitude and height of the meteor itself. The corresponding transformations are listed in the appendix. The procedure contains four steps: At first the geodetic coordinates of the radar (longitude-$\phi$,latitude-$\lambda$, height-$h$) are converted into ECEF, which means we receive a vector $\boldsymbol{x_R} = (X_R, Y_R, Z_R)$ with respect to the Earth center. From the interferometric solution we also know the position vector $\boldsymbol{x_P} = (x, y, z)$ with respect to the receiver location for each meteor. This vector $\boldsymbol{x_P}$

is then transformed into ECEF coordinates and we obtain a vector pointing from the Earth center towards the meteor position with coordinates $\boldsymbol{x_M} = (X_M, Y_M, Z_M)$. Further, we convert the vector $\boldsymbol{x_M}$ given in ECEF coordinates back into a geodetic position given by the latitude, longitude and height above the Earth surface for each meteor. Finally, we use the ECEF vector $\boldsymbol{x_M}$ and the geodetic reference to compute a local coordinate set ENU at the position of the meteor itself. A detailed description of all applied coordinate transformations is summarized in the appendix.

In summary we perform the following steps:

1. conversion of the geodetic coordinates of the radar $(\phi, \lambda, h) \rightarrow (X_R, Y_R, Z_R)$ by using the transformation geodetic to ECEF

2. transformation of meteor coordinates into ECEF $(x, y, z) \rightarrow (X_M, Y_M, Z_M)$ by using the transformation of ENU to ECEF

3. conversion of ECEF frame meteor position into geodetic coordinates $(X_R, Y_R, Z_R) \rightarrow (\phi, \lambda, h)$ ECEF to geodetic

4. determination of local ENU using the geodetic position of each meteor in ECEF coordinates $(\phi, \lambda, h) \rightarrow (x_M, y_M, z_M)$

Figure 2 shows an example of the difference between the ENU coordinates (black cross) of the radar location and the ENU coordinates (red cross) of a meteor observed at a horizontal distance of 300 km. The blue circle marks the 300 km range around the radar, which is assumed to be located at Juliusruh. Although the difference appears to be small, it introduces an error of a

few meters per second in the derived zonal and meridional and vertical wind speeds. Depending on the range and geographic latitude of the measurements the local azimuth and zenith shows differences up to 4 degrees compared to the radar site. As our wind measurements are supposed to be aligned along the zonal and meridional direction it is beneficial and straightforward to remove this bias. In the case of the standard SMR wind analysis technique (Hocking et al., 2001), where a homogenous wind is assumed within the observation volume, the error is almost compensated due to the large number of meteors used for the

wind estimate. However, it turns out that the random distribution of meteors within the observation volume is sometimes not favorable to compensate for this bias, thus, it also has an effect on the standard analysis. In particular, altitudes where only a small number of meteors are used for the wind estimation procedure are more prone to this type of error. In particular, it appears to be very critical or in fact almost impossible to obtain a reliable vertical wind velocity or momentum flux, if the full Earth geometry is not taken properly into account.





## 3.2 Retrieving arbitrary non-homogenous wind fields

The retrieval of arbitrary and non-homogenous wind fields is mathematically more demanding as the number of unknowns exceeds the number of measurements, which does not allow to directly solve the equations applying standard least squares or singular value decomposition algorithms. However, it is possible to constrain the problem by additional assumptions or a priori knowledge. Very often the smoothness is used to regularize the problem so that it can be solved by applying statistical inversion algorithms (Aster et al., 2013).

At first we define a spatial grid and a domain area. In the case of the German MMARIA network we use a $30 \times 30$km grid spacing in zonal and meridional direction. The total domain area is about $600 \times 600$km. The spatial grid is fixed for each altitude and follows the Earth's surface. A schematic view of the spatial grid is given in Figure 3. There are many other possibilities to define the spatial grid, e.g., using fixed longitude and latitude bins or arbitrary grids by using each individual meteor position. It turns out that spatial grids with fixed horizontal distance have benefits for the diagnostic of the wind fields as they allow to use discrete Fourier transforms or wavelength based spatial spectral analysis techniques. In this study, we have adopted a regular grid.

The first step of the wind field inversion procedure is to assign each observed meteor to a grid point $j$ centered at time $t$ and position $\boldsymbol{x_j}$. This is equivalent to averaging measurements in time and space. At present we use a 1 hour window shifted by 30 minutes and a vertical averaging kernel of 3 km centered at the respective altitude bin. In order to take into account that each observed meteor does not occur exactly at time $t$ and is not observed exactly at the position of the grid point $\boldsymbol{x_j}$, we assign a weight to each observation (Shepard, 1968), i.e.,

$$w_i(\boldsymbol{x_j}, \boldsymbol{x_i}) = \frac{\gamma_x}{|\boldsymbol{x_j} - \boldsymbol{x_i}|^p}. \tag{1}$$

The weight $w_i$ for meteor $i$ at position $\boldsymbol{x_i}$ and at time $t_i$ is inversely proportional to its distance to the center of the grid point $j$. The exponent $p$ is used to control how fast the weight is reduced as a function of distance. Assuming a value $p = 0$ results in a box car with equal weight for each meteor independent of its distance from the grid point. We use the distance in meters and $p = 0.2$. The term $\gamma$ is used to control the slope of the time distance – we use $\gamma_x = 1.0$ for time in units of seconds. The main reason for the averaging is that a single meteor, which lasts for 20-200 milliseconds, does not necessarily provide a representative mean total wind velocity for a 30 minutes time bin at a grid point. At least two meteors have to occur within one grid cell, otherwise we do not attempt to estimate the wind speed for that grid point. We will discuss later how this weight is applied in the inversion procedure.

We can relate the measured radial velocity of each meteor to the three dimensional wind velocity by using local ENU coordinates for each measurement $i$ as:

$$v_{r,i} = u_j \cos(\phi_i) \sin(\theta_i) + v_j \cos(\phi_i) \sin(\theta_i) + w_j \cos(\theta_i), \tag{2}$$

where $v_{r,i}$ is radial velocity for meteor $i$; $u_j, v_j$, and $w_j$ are the zonal, meridional, and vertical wind components at grid point $j$ corresponding to the meteor location. The angles $\theta_i$ and $\phi_i$ corresponding to meteor $i$ are the local ENU coordinates corresponding to the line of sight velocity along the direction vector from the radar. In the case of a forward scatter geometry





the radial velocity and the position of the 'radar' are more complicated and the radial velocity has to be corrected for the multi-static geometry (see (Stober and Chau, 2015)).

The radial wind equation for arbitrary measurements and grid points can be expressed as a linear matrix equation. The mapping from the zonal, meridional, and vertical components to observed radial velocities is given by a geometry matrix $G \in \mathbb{R}^{n \times 3m}$. All the measurements during an analysis interval are represented as a vector $v_r \in \mathrm{R}^{n \times 1}$, where $n$ is the number of measurements. The radial velocity vector $v_r$ contains all observed radial velocities, either for each individual meteor weighted

by its distance from the grid point or an averaged value that is already interpolated to the defined grid. The unknown 3D wind components at each grid point are also expressed as a vector

$$\boldsymbol{u} = [u_1, v_1, w_1, \cdots, u_m, v_m, w_m]^T \in \mathbb{R}^{3m \times 1}, \tag{3}$$

where $m$ is the number of grid points. The mapping in eq 2 can be compactly expressed using the following matrix equation:

$$\boldsymbol{v_r} = \boldsymbol{Gu}, \tag{4}$$

which relates all measured radial velocities to 3D velocities within a grid. More explicitly, this is

$$
\begin{bmatrix} \vdots \\ v_{r,i} \\ \vdots \end{bmatrix} =
\begin{bmatrix}
\ddots & \cdots & \cdots & \cdots & \vdots \\
\vdots & 0 & 0 & 0 & \vdots \\
\vdots & \cos(\phi_i)\sin(\theta_i) & \sin(\phi_i)\sin(\theta_i) & \cos(\theta_i) & \vdots \\
\vdots & 0 & 0 & 0 & \vdots \\
\vdots & \cdots & \cdots & \cdots & \ddots
\end{bmatrix}
\begin{bmatrix} \vdots \\ u_j \\ v_j \\ w_j \\ \vdots \end{bmatrix}. \tag{5}
$$

The geometry matrix $G$ combines all measurements from all possible viewing geometries, but it is not directly invertible. Although we have several different viewing geometries, we do not get always three independent measurements for each grid point. Hence, the number of unknowns is still larger than the number of measurements (rows in matrix $G$). This is in particular

the case at the edges of our domain area.

Ill-posed problems can be solved by adding additional constrains. Very often the smoothness of the unknown provides a reasonable way to regularize an ill-posed problem (Aster et al., 2013). The smoothness in our case corresponds to the spatial derivative for each wind component and grid point. This is equivalent to the assumption that wind field is only slightly changing between neighbored grid points. Hence, we define a smoothness matrix $L \in \mathbb{R}^{3m \times 3m}$ in such a way that we couple neighbored

grid points for each velocity component separately. For one velocity component, and one grid point, the elements of matrix $L$ would be:

$$
\boldsymbol{L} =
\begin{bmatrix}
\ddots & 0 & 0 & \cdots & 0 & \cdots & 0 & \cdots & 0 & 0 \\
\vdots & 4 & -1 & \cdots & -1 & \cdots & -1 & \cdots & -1 & \vdots \\
& 0 & 0 & \cdots & 0 & \cdots & 0 & \cdots & 0 & \ddots
\end{bmatrix} \tag{6}
$$





The matrix $\boldsymbol{L}$ contains such differences for all grid points and all velocity components.

Finally, we have to deal with grid points in the domain area where no measurement is available for a given time - altitude bin. This issue is solved by introducing a mesoscale wind field solution to these grid points. We tested three possible mesoscale solutions and checked how much the final solution depends on this mesoscale boundary condition. The most trivial way is zero padding or simply not using an explicit a priori for these points, the second one is estimating a mean wind using all radial velocity measurements and the third possibility is to derive a mesoscale wind field solution by computing local mean

winds for each multi-static geometry and to estimate a distance weighted background wind field for each grid point. A similar result is achieved by applying volume velocity processing (VVP) (e.g., Browning and Wexler, 1968; Waldteufel and Corbin, 1979), which was already successfully applied using horizontally resolved radial wind measurements (Stober et al., 2013) or multi-static SMR observations (Stober and Chau, 2015; Chau et al., 2017).

Combining all the information and the smoothness constraints into a set of equations allows to solve the ill-posed problem.

We obtain an estimate for the 3D wind components $\hat{\boldsymbol{u}}$ at all grid points solving the equation;

$$\hat{\boldsymbol{u}} = (\boldsymbol{G}^T \boldsymbol{\Sigma}^{-1} \boldsymbol{G} + \alpha \boldsymbol{L}^T \boldsymbol{L})^{-1} \boldsymbol{G}^T \boldsymbol{\Sigma}^{-1} \boldsymbol{v_r}, \tag{7}$$

which is a standard regularized weighted linear least-squares estimator (Aster et al., 2013). The matrix $\boldsymbol{\Sigma} = \mathrm{diag}(\sigma_1^2, \cdots, \sigma_n^2)$ is a diagonal matrix that contains the variance (i.e., measure of uncertainty) given to each measurement $\sigma_i^2$. The regularization parameter $\alpha$ provides a weight to the regularization constraint. It describes the coupling strength between neighboring grid

points. It should be noted that there also exists a number of alternatives to regularizing the solution.

When assigning the variance of each measurement $\sigma_i^2$, we use use the sample variance $\hat{\sigma}_i^2$, i.e., the variability of wind velocities of all measured wind velocities assigned to each grid point. The standard deviation ($\hat{\sigma}_i$) is typically in the order of a few cm/s up to several m/s. In addition to the natural measured variability of measured radial velocity, we also take into account the weight for each measurement, which is a function of distance of the measurement to the grid point. In other

words, we assume that because each measurement is not centered at the grid point center, there is an additional independent error that needs to be taken into account. Thus, $\sigma_i^2 = w_i(\boldsymbol{x_j}, \boldsymbol{x_i}, t, t_i)^{-2} + \hat{\sigma}_i^2$. There are grid points where a sufficient number of measurements are not available and the mesoscale solution is used. In this case, the measurement is weighted by a large uncertainty ($\sigma_i = 200$ m/s) to ensure that this does not strongly bias the inversion.

In the following we are going to demonstrate the robustness of our algorithm in dependence of the choice of regulariza-

tion parameter $\alpha$ and the mesoscale boundary conditions. For simplicity we will only focus on vertical winds, and leave the discussion of vertical mesoscale wind velocity for future work. In order to obtain reliable and physically meaningful vertical velocities, additional regularization constraints might be required.

### 3.3  Robustness of wind field solution

Solving eq. 7 is straightforward using singular value decomposition or matrix inversion algorithms. As the wind field inversion

is still a linear problem, we just need to find a proper solution for the regularization parameter $\alpha$. A large $\alpha$ means that our solution is dominated by the regularization constraint, a too small $\alpha$ results in a too weak coupling between neighboring grid





points, making the solution unstable. This is usually expressed in the so-called 'L-curve' (e.g., Aster et al., 2013).

In Figure 4 we compare the obtained wind fields for different strengths of the regularization parameters $\alpha$. The left picture shows what we consider an optimal solution with $\alpha = 0.014$. This optimal solution was estimated through several iterations. The wind field in the center was computed assuming a much too strong $\alpha = 10$. This obviously leads to a much too smooth wind field, but still keeps some mesoscale wind field structure. A much higher value in the order of 100 or 1000 is going to further reduce the shown variability. The right picture shows an example with an intentionally much to small regularization

constrain of $\alpha = 0.000001$. This obviously leads to some erratic structures and outliers begin to dominate the wind field. We tested different possibilities to define an optimal regularization strength $\alpha$. At present we optimize our solution with a global estimate that is valid for the whole domain, instead of estimating a local regularization strength $\alpha$ for each grid point. The local approach did not suppress erratic structures or outliers in the same way. After comparing thousands of images using different strengths and ways to estimate the optimal $\alpha$ it turned out that $\alpha = 0.1$ is very often a useful value, which leads to a

convergence within 8 iterations. However, the choice of $\alpha$ depends on the used statistical weights. As already mentioned above there are grid points where we have no direct measurements from one of the systems. We suggested to use a mesoscale solution for these points. Now there is the question whether our solution depends on this pre-described mesoscale wind field. Therefore we prepared two test cases. In the first one all grid points with no direct measurements are zero padded. For the second one we use a computed mesoscale wind field estimated from VVP. Figure 5 shows 4 pictures using the two test cases. Different

colors label grid points with direct radial velocity measurements (blue) and grid points with the mesoscale solution (red). The left plot displays the first iteration step and the right one shows the finally obtained wind field solution. North of 52° N there are almost no differences of the solution if one just compares the blue arrows. The main reason for this good agreement is that almost all points are linked by multiple observing geometries, whereas south of 52° N we basically have only monostatic observations. As a result the obtained wind field in the southern part of the domain area is more prone to be affected by the

boundary conditions. However, as there is almost no visible difference between the wind fields at latitudes north of 52° N, we conclude that there is almost no impact on the determined wind fields by the pre-described mesoscale winds.

## 4    Horizontally resolved wind fields and initial validation

The above described algorithm is applicable to all types of multi-static observations. In 2014 we started to build the MMARIA network in Germany. At present the network consists of 2 monostatic SMRs located at Juliusruh (54.6° N, 13.4° E) and Collm

(51.3° N, 13.0° E) (see Table 1). In addition to that we installed three receiver-only stations, namely a dual frequency station in Kuehlungsborn (54.1° N, 11.8° E) and a single frequency station in Juliusruh, i.e., resulting in 5 different links. Our preliminary results using such observations from two links (i.e., Jruh-Jruh, Kborn-Jruh) are described in detail in (Stober and Chau, 2015; Vierinen et al., 2016).

In parallel we also operated for one week a newly developed continuous wave (cw) coded system that complement our pulsed

SMR network. The CW-coded prototype was tested from 10 June 2015 until 12 June 2015 (Vierinen et al., 2016). The first campaign used the existing infrastructure by transmitting from Juliusruh and reception at Kuehlungsborn. From 14 March 2016





to 20 March 2016 there was a second CW-coded campaign where two temporary transmitters were installed. The transmitters were located in Luebs (53.7° N, 13.9° E) and Schwerin (53.6° N, 11.4° E), and operated at the same frequency as the Juliusruh pulsed system, i.e, 32.55 MHz.

During the March campaign in 2016 the multistatic network consisted of 2 monostatic and 5 multi-static links. Some technical specifications of the experiment settings of the SMRs and the locations of the multi-static links are summarized in Table 1 and 2, respectively. To simplify the discussion about the different viewing geometries we introduce the virtual radar location

of each multi-static link (see Figure 6). The derived Doppler velocities are measured with respect to the middle point of the corresponding transmitter and and receiver link, i.e. projected in the unit vector of the meteor position and this middle point. The resulting MR network is shown in Figure 7. The panel a) shows the position of all used transmitter and receiver sites. Panel b) shows the diversity of viewing geometries that results from the combination of the active radars and the multi-static links. The red points label either the position of the MRs or the virtual locations of the multi-static links.

In Figure 8 we show an overview of the zonal and meridional winds obtained from an all-sky fit as described above. The campaign was conducted during the transition from winter to summer circulation. The first three days are characterized by a mean westward zonal wind, which becomes weaker in the second half of the campaign, which is typical for this time of the year (e.g., Wilhelm et al., 2017). The mean meridional wind is close to zero. Both wind components indicate a moderate tidal activity, which is dominated by the semi-diurnal tide with a tidal amplitude of less than 50 m/s.

For the same period we generated three movie sequences at 82, 90, 96 km altitude. They show the 2D wind fields and their temporal evolution with 1 hour time steps. The movies can be found in the supplementary material. The clockwise rotation of vector field is mainly due to the semi-diurnal tide. However, the movies also show the temporal and spatial variability due to GWs. The appearance and disappearance of the red points indicates whether this viewing geometry was available during the inversion or not. Note that arrows are scaled within each image. As a result more distorted wind fields are often related to weak

winds (<20 m/s), whereas very smooth wind fields are often related to higher wind speeds (>40 m/s).

Figure 9 presents two examples of obtained 2D wind fields. The domain area was reduced and optimized to cover the Baltic coast where most of multi-static meteor links are concentrated. Further the retrieval used here did not make use of a explicit mesoscale wind field regularization. The left figure indicates a potential body force of a breaking GW causing an acceleration of the flow (Vadas and Fritts, 2001). The second example indicates a closed small scale vortex above the Baltic coast. The

vortex is also rather likely the result of a body force event in the North East corner of the domain area accelerating the flow towards the south west direction.

We did also perform an initial validation of the derived wind field for the complete campaign period through testing the consistency of the wind observations compared to the all-sky fit and the VVP. A comparison of the mean zonal and meridional winds obtained from the all-sky fit and the mean wind velocities over the domain area and for all available altitudes between

82-98 km are shown in Figure 10. The mean wind velocity was obtained by summation of all grid points that are constrained by observations. The comparison shows that there is a remarkable agreement between the all-sky fit and the mean 2D wind velocity within the domain area. The correlation is as high as 0.98 for the zonal mean wind and 0.97 for the mean meridional winds. The slightly weaker agreement of the meridional component is likely related to the in general lower meridional wind





speeds compared to the zonal winds.

We also performed an initial comparison between the VVP derived wind estimates for each grid point and the 2D horizontally resolved wind fields. We compared all wind velocities at all grid points between 82-98 km altitude. In Figure 11 we show the resulting correlation density plot. The correlation of the zonal and meridional winds are lower compared to the agreement

of the mean winds. The main reason for the increased scattering is attributed to two effects. Firstly, the VVP uses a linear extrapolation towards the edges of the domain area, which is not necessarily the best approximation. Secondly, the 2D wind retrieval reveals much more of the small scale structures in the wind field compared to the VVP.

## 5 Obtaining horizontal wavelength spectra

The new wind retrieval algorithm opens new possibilities to investigate atmospheric dynamics. The spatial information seems

to be useful to study the horizontal wavelength/wavenumber power spectra of kinetic energy. For the troposphere and lower stratosphere, Nastrom and Gage (1985) analyzed about 6000 commercial aircraft flights. They found a spectral slope of $k^{-5/3}$ for wavelengths between 2.6 and 400 km, which steepens to $k^{-3}$ for larger wavelengths. The $k^{-5/3}$ slope is considered to be representative for mesoscale GWs, whereas the steeper slope is more characteristic for the synoptic scale.

Due to the regular spatial grid horizontal wavelength spectra are easily obtained from the derived horizontally resolved wind

fields. The mean spectrum is computed by adding all latitudinal cuts through the domain area at a given altitude during one day. Considering that the coverage of our 2D wind fields is variable, we included only latitudinal cuts with more than 12 grid points constrained by measurements. The resultant spectra are shown in Figure 12. The grey points are obtained by plotting each individual spectrum. The black line indicates the mean spectrum. We further added two reference slopes with $k^{-5/3}$ and $k^{-3}$. The straight green and black lines are estimated by a linear fit and label two different slopes using a wavelength window

60-140 km (green) and 140-800 km (black). The vertical light blue lines represent the domain boundaries. The spectra are estimated by using Lomb-Scargle periodograms (Lomb, 1976; Scargle, 1982).

The spectra shown in Figure 12 suggests that our domain area has a sufficient large extension to get an idea of the transition between the $k^{-5/3}$ to $k^{-3}$ spectral slopes at the mesosphere. However, we have not yet gathered enough statistics to pinpoint a transition scale. Typically, the synoptic scale is associated to a more vortical driven flow, whereas the mesoscale GW flow

regime is characterized by divergent modes or GWs. At least for the 5 campaign days there is only a weak day-to-day variability.

## 6 Discussion

Retrieving horizontally resolved wind fields from multi-static SMR networks at the MLT provides new possibilities to investigate the intermittency and spatial characteristics of GWs and vortical modes, which are not yet accessible by other remote sensing techniques for these spatial scales and with that temporal resolution. In particular, the wind and its spatial charac-

teristics are required to understand wave breaking and the associated momentum transfer to the background (e.g., Fritts and Alexander, 2003; Fritts et al., 2014).





A crucial part of the presented wind field analysis is the spatio-temporal sampling. Increasing the spatial resolution is only meaningful if we also decrease the temporal sampling window. However, with a decreasing number of detections within the domain area the more sparse is our wind field constrained. This brings us to the question on how representative is the observed radial velocity of an individual meteor for our selected time, altitude and spatial resolution. If we want to resolve small scale structures with characteristic life times of minutes and horizontal scales comparable to our grid resolution, e.g., bore events or breaking GW (ripples) (Hecht et al., 2007), a much denser MR network is desirable. However, even with the present

stage of the MMARIA Germany network we are able to resolve mesoscale vortical modes as well as GWs. As a result of the spatio-temporal sampling we expect to be more sensitive to vortical modes and to resolve the effects of body forces of breaking GW (Vadas and Fritts, 2001). The obtained 2D wind fields are also ideal to complement other mesospheric measurements. The combination of the horizontally resolved wind fields with other mesospheric observations like airglow imagers (e.g. Hecht et al., 2007) or the Advanced Mesospheric Temperature Mapper (Taylor et al., 2009; Pautet et al., 2014, 2016) is going to

provide a more complete picture of the MLT dynamics. The horizontally resolved wind fields allow a better characterization of the mesoscale mean flow. The airglow imagers provide more information to small scale structures (a few kilometers).

During the past years there were also several attempts to retrieve horizontally resolved wind fields using Fabry-Perot Interferometer (FPI) in the thermosphere (Meriwether et al., 2008; Harding et al., 2015). In particular, Harding et al. (2015) used a comparable retrieval and constrained the wind field solution by the smoothness and the curvature. They showed data collected

above Brazil using up to 7 FPI combined to a FPI network. They obtained rather smooth wind fields similar to those shown in Figure 4 ($\alpha = 10$). A combination of a FPI and a MR network in combination with further optimized retrieval algorithms can enhance our understanding on the vertical coupling between the layers and the propagation of waves and their interaction and dissipation.

## 7  Conclusions

After establishing the MMARIA-concept in Stober and Chau (2015), we extended the SMR network in Germany, which now consists of two monostatic SMRs at Juliusruh and Collm and three multi-static links between Juliusruh-Kborn, Collm-Kborn and Collm-Juliusruh. Further, we investigated new technological concepts by adding two CW-coded transmitter (Vierinen et al., 2016). Here we present initial results from a 5 day campaign in March 2016 combining pulsed and cw-coded multi-static SMR observations, that resulted in 7 links.

The introduced wind retrieval algorithm for arbitrary non-homogeneous wind fields shows the potential to investigate mesoscale dynamics at the MLT by employing multi-static SMR networks. Horizontally resolved winds open possibilities to study the MLT dynamics. We demonstrate that our preliminary derived wind fields are suitable to obtain horizontal wavelength spectra to access the transition scale between a $k^{-3}$ to a $k^{-5/3}$ slope, that has been observed in the troposphere (Nastrom and Gage, 1985).

We performed an initial validation of our wind retrieval algorithm by comparing the mean winds to the standard SMR wind analysis, which shows a remarkable good agreement. Further, we compared the wind fields obtained from VVP, using a gra-





dient extrapolation of the winds to our grid points with our 2D wind retrieval solution. This comparison reveals that both methods provide a good approximation of the mesoscale wind field, but show larger discrepancies at the smaller scales, which is expected as the 2D wind retrieval of 3D wind vectors is designed to infer such small scale features.

The presented algorithm demonstrates the potential of SMRs to be used as networks. These systems are cheap enough and

sufficiently automated to be deployed at remote locations and to build rather large networks with several hundred kilometers in diameter. Further, the derived wind retrieval algorithm is applicable to existing data collected from multi-SMRs like in Scandinavia (Chau et al., 2017).

*Competing interests.*  There are no competing interests.

*Acknowledgements.*  We acknowledge the technical support of our radar systems by Falk Kaiser, Dieter Keuer, Jörg Trautner and Thomas

Barth. We are grateful for the support provided by S. Geese, N. Pfeffer and M. Claßen, J. Trautner, in developing, deploying, operating, and analysizing the CW-coded radars.



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





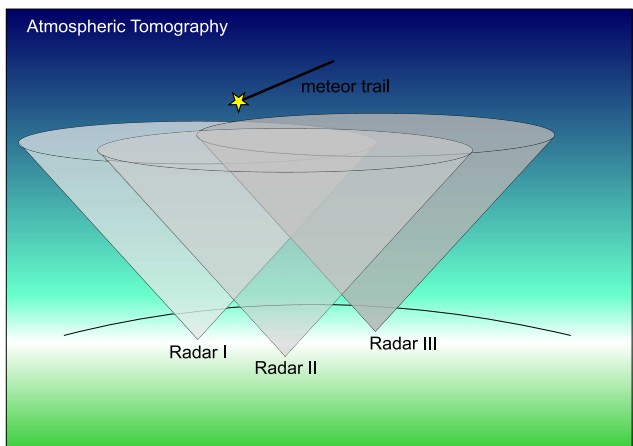

**Figure 1.** Schematic of a multi-static meteor radar network. The grey shaded areas stand for the typical field of views for each systems. Within the network all system should at least overlap to one of the other network members.

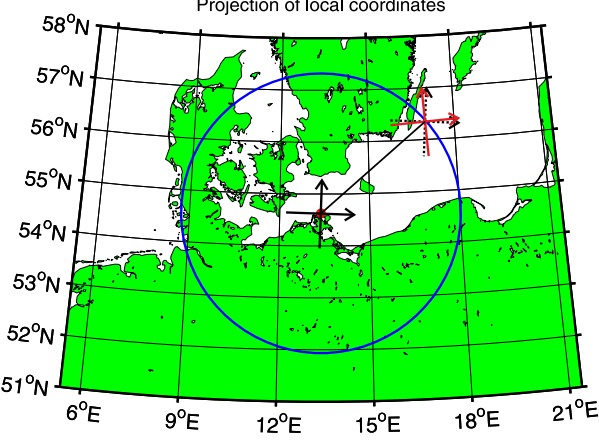

**Figure 2.** Illustration on how local coordinates (zonal, meridional) change with geographic position with respect to a radar location.





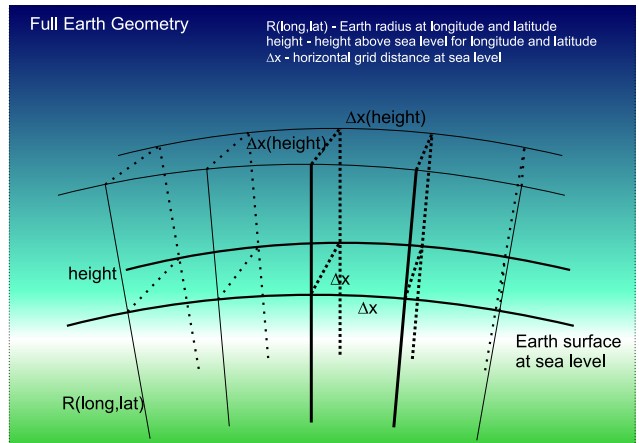

**Figure 3.** Schematic of 3D gridding to compute horizontally resolved wind fields including the Earth surface.

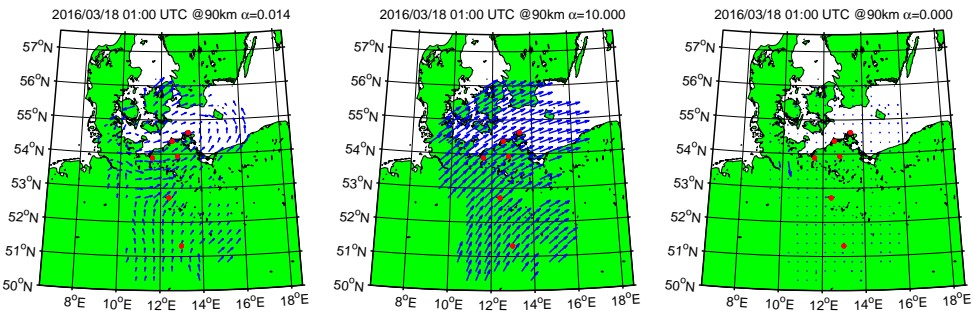

**Figure 4.** Comparison of 2D wind fields for three different $\alpha = 0.014$, $\alpha = 10$ and $\alpha = 0.000001$ The length of the arrows between the images does not scale.

**Table 1.** Technical specifications of the two active meteor radars.

| parameter | Juliusruh | Collm |
|---|---|---|
| frequency (MHz) | 32.55 | 36.2 |
| power Tx (kW) | 30 | 15 |
| PRF (Hz) | 625 | 625 |
| range resolution (km) | 1.5 | 1.5 |
| antenna | crossed dipole | crossed dipole |
| operation | pulsed | pulsed |
| Code | 7-bit Barker | 7-bit Barker |
| location | 54.6° N, 13.4° E | 51.3° N, 13.0° E |

PRF - pulse repetition frequency





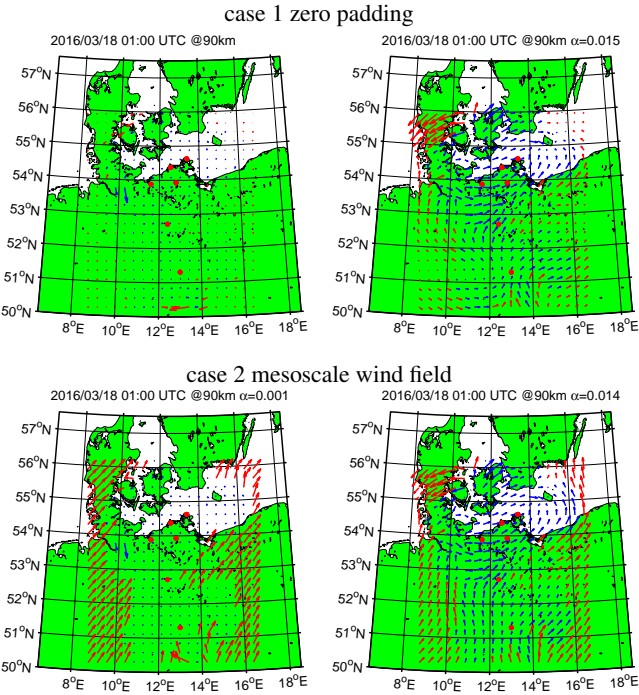

**Figure 5.** Comparison of wind field solution in dependence of the background mesoscale solution. upper panels: test case with zero padding first (left) and final iteration (right). lower panels: test case with mesoscale wind field (obtained from the data) first (left) and final iteration (right).

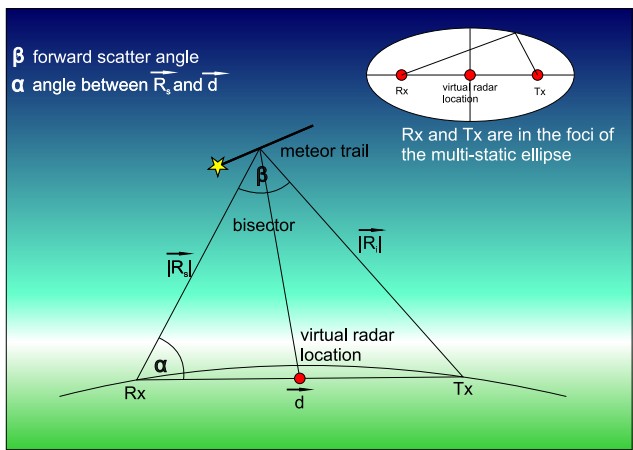

**Figure 6.** Schematic of a typical forward scatter meteor radar. The position of the Tx and Rx are known and all other parts are measured. The Bragg vector $k_b$ always points towards the center of the direction vector $d$.





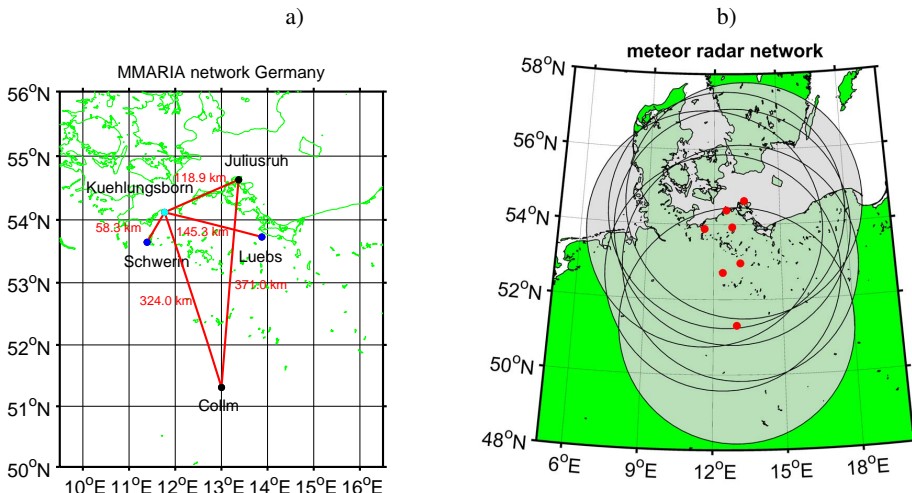

**Figure 7.** a) Schematic view of multi-static network during the campaign in March 2016. b) Geographic map of different viewing geometries (red points). The shaded areas labal a circle of 300 km in diameter around each center of radial velocity measurements.

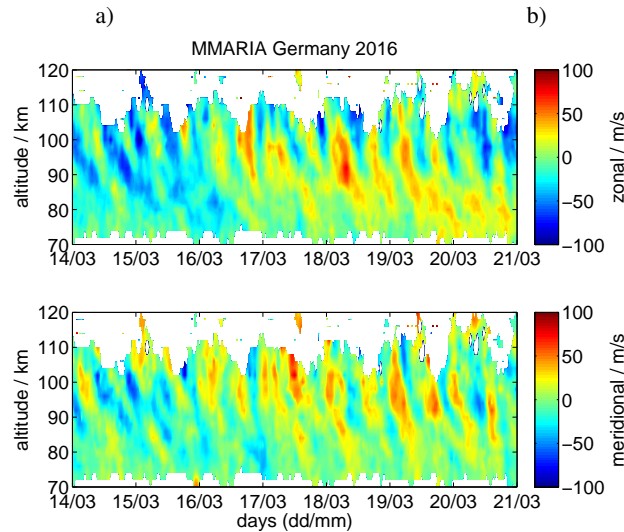

**Figure 8.** Overview of of zonal and meridional wind components applying the standard mean wind analysis to the MMARIA network during the March 2016 campaign.





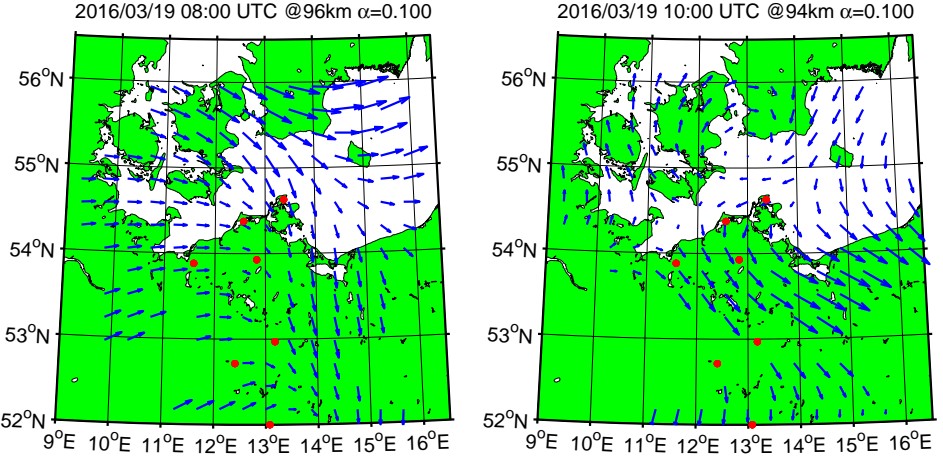

**Figure 9.** Two examples of obtained wind fields showing a small vortical structure above the Baltic coast and a potential body force of a breaking GW.

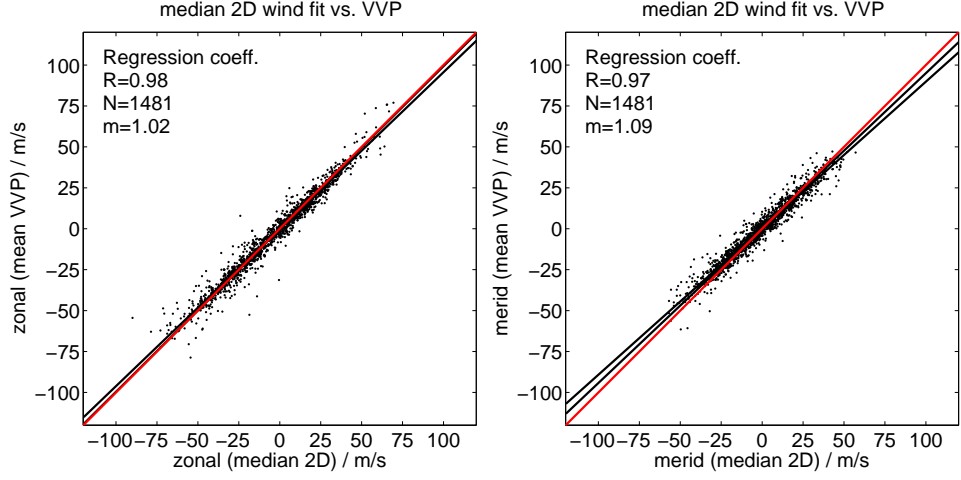

**Figure 10.** Scatter plots of mean zonal and meridional winds obtained from the 2D regularized wind field fitting and VVP. The mean is computed as average above the domain area.





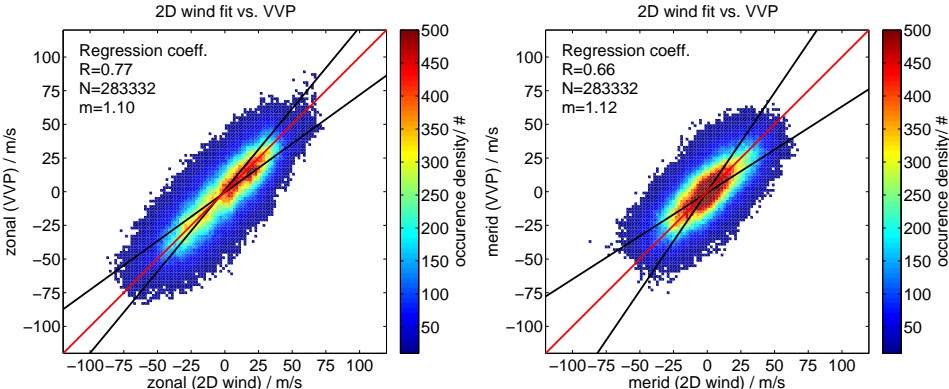

**Figure 11.** Comparison of zonal and meridional winds as derived from the new retrieval algorithm and the estimates for each grid point applying VVP.

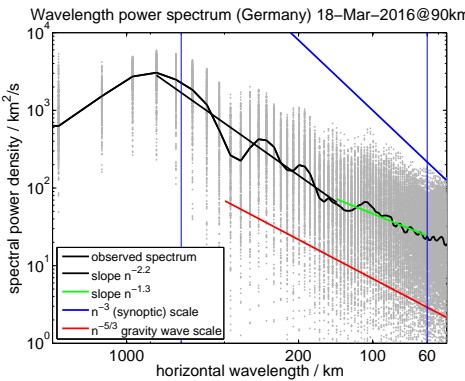

**Figure 12.** Horizontal wavelength spectra and estimated slopes to identify the transition from the mesoscale GW toi the synoptic scale.

**Table 2.** Technical specification of the multi-static links used in the experiment campaign in March 2016.

| parameter | Juliusruh-Kborn | Collm-Kborn | Collm -Juliusruh | Luebs - Kborn | Schwerin-Kborn |
|---|---|---|---|---|---|
| location Tx | 54.6° N, 13.4° E | 51.3° N, 13.0° E | 51.3° N, 13.0° E | 53.7° N, 13.9° E | 53.6° N, 11.4° E |
| location Rx | 54.1° N, 11.8° E | 54.1° N, 11.8° E | 54.6° N, 13.4° E | 54.1° N, 11.8° E | 54.1° N, 11.8° E |
| virtual location | 54.4° N, 12.6° E | 52.7° N, 12.4° E | 53.0° N, 13.1° E | 53.9° N, 12.8° E | 53.9° N, 11.6° E |
| frequency (MHz) | 32.55 | 36.2 | 36.2 | 32.55 | 32.55 |
| operation principle | pulsed | pulsed | pulsed | cw | cw |
| transmitted power | | | | 500 W | 500 W |
| distance (km) | 118.6 | 323.6 | 370.6 | 144.8 | 58.2 |

Kborn-Kuehlungsborn



**Appendix A: Coordinate transformations**

In the following we present a short summary of all the coordinate transformations that we used in the presented analysis scheme. All relevant parameters are listed and the used transformation matrices are shown.

15 **A1 Geodetic to ECEF**

The first transformation that we used converts geodetic coordinates into the ECEF. The geodetic coordinates from the radar are given in longitude ($\lambda$), latitude ($\phi$) and height ($z$). The height denotes the altitude of the radar above Earth surface with respect to WGS84 (National Imagery and Mapping Agency, 2000; Hofmann-Wellenhof et al., 1994). The WGS84 defines the semi-major axis of the Earth to be $a = 6387137.0m$ and a reciprocal of flattening of $f = 1/298.257223563$. The semi-minor

20 axis is defined to be $b = 6356752.3142m$. The first eccentricity squared $e^2$, the second eccentricity squared $e'^2$ and the radius of Earth's curvature $N$ is given by

$$
\begin{aligned}
e^2 &= 2 \cdot f - f^2 \ , \\
e'^2 &= (a^2 - b^2)/b^2 \ , \\
N &= a\sqrt{(1 - e^2 \sin(\phi)^2)} \ .
\end{aligned}
\tag{A1}
$$

Based on the WGS84 ellipsoid geometry of the Earth any given geodetic location defined by a longitude, latitude and a height (height above WGS84 surface) is given by the ECEF coordinates $(X_R, Y_R, Z_R)$

$$
\begin{aligned}
X_R &= (N + z) \cdot \cos(\phi)\cos(\lambda) \ , \\
Y_R &= (N + z) \cdot \cos(\phi)\sin(\lambda) \ , \\
Z_R &= (N + z - e^2 N)\sin(\phi) \ .
\end{aligned}
\tag{A2}
$$

**A2 ECEF to Geodetic**

The backward transformation to transform a given coordinate in ECEF into a geodetic longitude ($\lambda$), latitude ($\phi$) and height ($z$) is more difficult. A summary of possible algorithms is provided in Zhu (1993). We apply the closed form presented in

10 Heikkinnen (1982). According to (Zhu, 1993) the average error is mainly determined by the numerical error introduced in the computer, which is in the order of 1 nm. The algorithm presented in Heikkinnen (1982) is valid from the Earth center (z=-6300 km) up to height of geostationary orbits.





$$
\begin{aligned}
e'^2 &= (a^2 - b^2)/b^2 & \text{(A3)} \\
F &= 54b^2 z^2 \\
G &= r^2 + (1 - e^2)z^2 - e^2(a^2 - b^2) \\
c &= e^4 F r^2 / G^3 \\
s &= \sqrt[3]{1 + c + \sqrt{c^2 + 2c}} \\
P &= \frac{F}{3(s + 1/s + 1)^2 G^2} \\
Q &= \sqrt{1 + 2e^4 P} \\
r_0 &= -\frac{Pe^2 r}{1 + Q} + \sqrt{\frac{a^2}{2}\left(1 + \frac{1}{Q}\right) - \frac{P(1 - e^2)z^2}{Q(1 + Q)} - \frac{Pr^2}{2}} \\
U &= \sqrt{(r - e^2 r_0)^2 + z^2} \\
V &= \sqrt{(r - e^2 r_0)^2 + (1 - e^2)z^2} \\
z_0 &= \frac{b^2 z}{aV} \\
h &= U\left(1 - \frac{b^2}{aV}\right) \\
\phi &= \arctan((z + e'^2 z_0)/r) \\
\lambda &= 2\arctan\left(\frac{\sqrt{X_R^2 + Y_R^2} - X_R}{Y_R}\right)
\end{aligned}
$$

**A3  ENU to ECEF**

Typically, MR observe meteors at a given distance and direction relative to its geodetic coordinates. The meteor is given in ENU coordinates with respect to the radar location. The up direction is defined by the tangent plane to the Earth's ellipsoid. The meteor position is defined by ENU coordinates $(x_m, y_m, z_m)$ at a geodetic longitude ($\lambda$), latitude ($\phi$) and height ($z$). Hence, we have to rotate the ENU-vector $(x_m, y_m, z_m)$ into the ECEF reference $(X_m, Y_m, Z_m)$ system by using the following expression;

$$
\begin{bmatrix} X_m \\ Y_m \\ Z_m \end{bmatrix} = \begin{bmatrix} -\sin(\lambda) & -\sin(\phi)\cos(\lambda) & \cos(\phi)\cos(\lambda) \\ \cos(\lambda) & -\sin(\phi)\sin(\lambda) & \cos(\phi)\sin(\lambda) \\ 0 & \cos(\phi) & \sin(\phi) \end{bmatrix} \cdot \begin{bmatrix} x_m \\ y_m \\ z_m \end{bmatrix} + \begin{bmatrix} X_r \\ Y_r \\ Z_r \end{bmatrix} . \tag{A4}
$$

**A4  ECEF to ENU**

Finally, we want to express our line of sight vector from the radar towards the meteor in the frame of the ENU coordinates at the geodetic location of the meteor itself, e.g., the line of sight velocity vector is observed at a certain azimuth $az$ and zenith $ze$ angle relative to the radar, but has a different azimuth $az'$ and zenith $ze'$ in the frame of the local geodetic coordinates of the meteor. Assuming that we know the ECEF coordinates of the meteor $(X_m, Y_m, Z_m)$ and our radar location $(X_R, Y_R, Z_R)$



it is straightforward to compute the ENU $(x, y, z)$ by using;

$$\begin{bmatrix} x \\ y \\ z \end{bmatrix} = \begin{bmatrix} -\sin(\lambda) & \cos(\lambda) & 0 \\ -\sin(\phi)\cos(\lambda) & -\sin(\phi)\sin(\lambda) & \cos(\phi) \\ \cos(\phi)\cos(\lambda) & \cos(\phi)\sin(\lambda) & \sin(\phi) \end{bmatrix} \cdot \begin{bmatrix} X_m - X_R \\ Y_m - Y_R \\ Z_m - Z_R \end{bmatrix} \quad . \tag{A5}$$

525    Hence, we obtain a local azimuth $az'$ and $ze'$ with respect to the geodetic position of the meteor, instead of the radar site;

$$az' = \quad \arctan(y/x) \tag{A6}$$
$$ze' = \quad \arccos\left(\frac{z}{\sqrt{x^2 + y^2 + z^2}}\right) \quad .$$