# Peer review of "Retrieving horizontally resolved wind fields using multi-static meteor radar observations"

_Atmospheric Measurement Techniques, 2018_

## Referee Comment (RC1) · Anonymous Referee #1 · 25 May 2018

The manuscript entitled 'Retrieving horizontally resolved wind fields using multi-static meteor radar observations' by Stober et al. demonstrates the potential of multi-static VHF radar networks for the determination of the 3-dimensional wind field in the mesosphere / lower thermosphere, when combined with appropriate retrieval schemes.

The work presented here is sound and well described. It is well suited for the journal and interesting to the community.

I suggest to clarify a few points regarding the retrieval methodology and the work logic before final publication:

page 5, line 11: Is an observed meteor assigned to a grid point or a grid cell? I assume the latter, such that it is assigned to multiple grid points, using the weighting mentioned

in equation (1)? Pls clarify.

page 5 line 12: What do you mean by '1 hour window shifted by 30 minutes'?

page 5, line 13: 'vertical averaging kernel' (avk) is commonly used as a diagnostic quantity in 1-dim constrained retrieval schemes. It cannot be 'used'. Is the regularization chosen in such a way that the avk has a halfwidth of 3 km? Or does this refer to the grid cell spacing in the vertical? Pls clarify. I also like to see an avk plot for a few typical cases

page 5, line 20: Pls justify the choice of p (=0.2). Why not 1?

page 6, equation 6: It is difficult to understand the logic behind this representation. Pls specify L in another way and explain the logic behind. Is the regularization the same for all grid points?

page 7, line 3: How is this considered within the inversion algorithm in detail?

page 7, line 4: Did you consider to adapt the regularization matrix in such a way that the regularization is much stronger (and more extended in space or time) for those cells, which do not coincide with a meteor trail? This is a common technique in other applications using inverse modeling.

page 7, line 13: How is the variance (of a single measurement?) defined? Or is it the variance of all measurements within a grid cell? It is common that sigma_i considers other sources of uncertainty as well, in addition to pure statistical measures. Please specify it in more detail.

page 7, line 21: How is the measurement space weighting factor sigma_i considered in the inversion formalism, if it depends on the position of the measurement and the position of the unknown(s)? What is a 'sufficient' number of unknowns?

page 9, line 4. I suggest to explain 'virtual radar location' in the text instead of in the figure caption.

page 11, line 4: Please be more quantitative here. Do you need more stations in the given volume/ area? What would be the gain in spatial resolution, if, e.g., the number of stations would be doubled?

page 19, Fig. 5: A difference plot for the two final iterations would be good to assess the differences of the two approaches.

———————————————————

---

## Referee Comment (RC2) · Anonymous Referee #2 · 21 Jun 2018

General comments:

The manuscript has two major themes regarding horizontal wind retrieval, namely: 1) the use of multi-static meteor radar observations giving both an increased wind field area compared to single specular meteor radars (SMR) as well as more meteor trail measurements from the volume, and 2) the use of regularization for retrieval of the horizontal wind velocities and improving the spatio-temporal resolution. The first objective required mapping the radar observations to the geoid of the Earth ('full Earth geometry') to properly describe the now increased wind field coverage area. The latter objective allows for smaller scale studies previously not possible with SMRs, such as gravity wave studies.

The manuscript achieves these objectives and is of significant scientific interest to the

field. This manuscript is a scientific advancement definitely of publication quality. However, there are some points which need clarification to better present the significance of this research.

Specific comments:

P.2, lines 8-18 (2nd paragraph) and Section 2: The new application of regularization to fitting horizontal winds is to be compared to: 1) the 'normal' or 'standard' (all-sky?) meteor radar wind retrieval technique, and 2) volume velocity processing (VVP) technique. Section 2 is to summarize the 'normal' or 'standard' meteor radar wind retrieval process while there is no summary of the VVP technique. Clarity of this over arching objective would be greatly benefitted if both the 'normal' and VVP techniques were summarized in Section 2 and contrasted with the new regularization method (obviously with the in-depth details of regularization presented in Section 3). That is, what are the differences between the fitting methods and what will the new regularization method add.

P.3, lines 13-16: What is a typical number of meteor trails which gives statistical uncertainty of 1-6 m/s at altitudes between 82 and 95 km?

P.4, l.21: This is the first time azimuth and zenith are mentioned. The azimuth and zenith (elevation) angles, along with angle of arrival should be defined earlier with respect to the defined axes systems of: a) the radar(s)/links, and b) the local co-ordinates.

P.5, l.20: How are values for p and $\gamma_x$ determined for Equ. 1?

P.6, l.22, Equ. 6: The terms defining the spatial derivative, time derivative?, etc. in the smoothness matrix L are unclear. Please define and clarify.

P7, l.16: For the variance $\sigma_i^2$, why not just use the measurement error?

P8, l.3: What do the authors mean by "optimal solution" for the regularization parameter $\alpha$? And how was the regularization parameter $\alpha$ value obtained/ justified by being "estimated through several iterations"? Also, this optimal value of 0.014 is not

the global value typically used (which is \alpha = 0.1). Why is this?

P8, lines 3-5, Fig. 4: I like how the authors used extreme values of the regularization parameter \alpha to show different fits, but I have some concern on how different the fits appear. Although this may mostly be due to the winds in the plot on the right, with \alpha = 10ˆ{-6}, not being scaled between the images. Why is this? And can they be re-scaled. Explain and clarify.

Also, for the \alpha = 10ˆ{-6} case, is this weak regularization essentially the 'normal' or 'standard' fitting method as \alpha goes to zero? Once again, this relates to comparing the regularization technique to the other two fittings techniques ('normal' and VVP).

P8, l.10, supplementary movies: Should not the regularization parameter \alpha be the same for all fits? Please justify and explain selecting different values of the regularization parameter.

P9, l.10, Fig. 8: One would assume that the "all-sky fit as described above" and then presented in Fig. 8 would be the new regularization technique, but according to the Fig. 8 caption it is the 'standard' mean wind analysis. If this is the case, why is the new regularization technique not used?

Then on P9, l.28 the "all-sky fit" clearly refers to the 'normal' or 'standard' fit. Again, please clarify and standardize terminology to different fitting methods.

P.9, l.27 to P.10, l.7: This text validates that the mean 2D wind fit (regularization fit I assume) agrees well with previous accepted fitting techniques (all-sky or 'standard' and VVP fits). If the new regularization is the same as the accepted fits, what has been gained by this new technique. This should be related back to Section 2 and the benefits of using regularization should be elaborated.

P.10 Discussion Section, P.11, lines 25-29: Are there any other benefits to the small-scale structures that are detectable using the new regularization scheme besides the behavior of the GW spectral slope? If so, list a few.

Technical corrections:

There are a number of grammatical errors, but these will be corrected in the copy editing stage.

P.5, l.27, Equ. 2: For 2nd term on RHS of equation it should be \sin(\phi_i) ˆˆˆ P.7, l.25: Do you mean "focus on horizontal winds", not "vertical" winds?

––––––––––––––––––––––––––––––

---

## Author Comment (AC1) · 17 Jul 2018

The authors thank the reviewer for this suggestions and comments about the paper. We followed the suggestions and modified or added the requested information. The changes are labelled in the revised version. Some paragraphs are expanded, as this was suggested by the second reviewer. We attached a pdf with all the changes that we included in manuscript generated by difflatex.

Comment: page 5, line 11: Is an observed meteor assigned to a grid point or a grid cell? I assume the latter, such that it is assigned to multiple grid points, using the weighting mentioned in equation (1)? Pls clarify.

Reply: Correct. The term grid cell is more adequate to describe the procedure. We

changed it throughout the manuscript.

Comment: page 5 line 12: What do you mean by '1 hour window shifted by 30 minutes'?

Reply: We rephrased this passage. Usually the data is analyzed using some over-sampling. That means a 30 minutes temporal resolution is achieved with an one hour window. We now changed this by introducing the Gaussian weights as equation. The vertical resolution is truncated plus/ minus 1.5 km above the reference altitude.

Comment: page 5, line 13: 'vertical averaging kernel' (avk) is commonly used as a diagnostic quantity in 1-dim constrained retrieval schemes. It cannot be 'used'. Is the regularization chosen in such a way that the avk has a halfwidth of 3 km? Or does this refer to the grid cell spacing in the vertical? Pls clarify. I also like to see an avk plot for a few typical cases

Reply: Thanks for pointing at the not optimal explanation. We use a Gaussian weight within a certain time and altitude bin. It is not the same as the averaging kernel in retrievals. The weights are estimated depending on the total shear, thus, are not constant. This Gaussian weighting seems to be usable to account for the randomness in the temporal and spatial sampling and add to the total error balance.

Comment: page 5, line 20: Pls justify the choice of p (=0.2). Why not 1?

Reply: We added a sentence explaining our reasoning behind the value of 0.2. There was not yet a deeper investigation whether this can be further optimized. As we measure the distance in m, a value of p=1, would lead to a vanishing impact of measurements at the edge of each cell. So the value of 0.2 ensures that even the meteors are between two grid cells have still a reduced but non-negligible impact.

Comment: page 6, equation 6: It is difficult to understand the logic behind this representation. Pls specify L in another way and explain the logic behind. Is the regularization the same for all grid points?

Reply: Indeed, is the representation of the smoothness matrix not easy. We added a scheme explaining the ratio behind our smoothness matrix. The L matrix contains entries for each wind component. Basically it is a diagonal matrix with filled side diagonals for each component. At the edges the matrix looks different, as the number or neighbor grid cell decreases. The matrix is set up as a new set of linear equations minimizing the difference between the selected grid cells for each component.

Comment: page 7, line 3: How is this considered within the inversion algorithm in detail?

Reply: We added a short sentence how the mesoscale constrain enters the retrieval. We set up a matrix with diagonal elements pre-describing a solution for this grid cell weighted by the regularization strength.

Comment: page 7, line 4: Did you consider to adapt the regularization matrix in such a way that the regularization is much stronger (and more extended in space or time) for those cells, which do not coincide with a meteor trail? This is a common technique in other applications using inverse modeling.

Reply: We thank the reviewer for this suggestion. Indeed, we did conduct some very basic tests with 'local' or 'regional' varying regularization strengths, but did not yet integrate them in the analysis. As we are going to increase the number of system with time and thus domain area such regional dependent regularization strengths seem to be useful.

Comment: page 7, line 13: How is the variance (of a single measurement?) defined? Or is it the variance of all measurements within a grid cell? It is common that sigma_i considers other sources of uncertainty as well, in addition to pure statistical measures. Please specify it in more detail.

Reply: We rephrased this part to avoid misunderstandings. We use two types of retrievals- on is now called full wind retrieval were each meteor is used with statistical

uncertainties in the radial velocity, angles, and wind components. The other retrieval is called 'packed' wind retrieva. In this case we compute a total wind velocity variance for each grid cell as weight. The errors due to angles and the wind components are treated the same way. Thus, the sigma_i is always a total error budget considering many different sources of errors.

Comment: page 7, line 21: How is the measurement space weighting factor sigma_i considered in the inversion formalism, if it depends on the position of the measurement and the position of the unknown(s)? What is a 'sufficient' number of unknowns?

Reply: The spatial weighting is just one quantity of many in the total error budget for each grid cell. In the packed wind retrieval at least 2 meteors are required. The full wind retrieval just requires one meteor per grid cell.

Comment: page 9, line 4. I suggest to explain 'virtual radar location' in the text instead of in the figure caption.

Reply: The term virtual radar location is explained in the text (page 10 line 09-20).

Comment: page 11, line 4: Please be more quantitative here. Do you need more stations in the given volume/ area? What would be the gain in spatial resolution, if, e.g., the number of stations would be doubled?

Reply: As suggested by the reviewer, we included some explicit numbers of how the meteor counts would affect the analysis. At present an increase would mainly be used to increase the temporal resolution. We aim to achieve 10-15 minutes as soon as our new planned stations are going to become available. An increase of the spatial resolution requires maybe a factor 4-8 more meteor counts.

Comment: page 19, Fig. 5: A difference plot for the two final iterations would be good to assess the differences of the two approaches.

Reply: We introduced two names for the different weighting approaches and compare them as sequence as well as with a scatter density plot.

Please also note the supplement to this comment:
https://www.atmos-meas-tech-discuss.net/amt-2018-93/amt-2018-93-AC1-supplement.pdf

[Figure]

**Supplement:**

[revised manuscript text omitted]

$$

---

## Author Comment (AC2) · 17 Jul 2018

The authors thank the reviewer for reading the manuscript and his suggestions. We updated the manuscript accordingly and the changes are labelled with the track changes in latexdiff. We are grateful for the provided review also spotting our mistake in the attached videos.

Comment: P.2, lines 8-18 (2nd paragraph) and Section 2: The new application of regularization to fitting horizontal winds is to be compared to: 1) the 'normal' or 'standard' (all-sky?) meteor radar wind retrieval technique, and 2) volume velocity processing (VVP) technique. Section 2 is to summarize the 'normal' or 'standard' meteor radar wind retrieval process while there is no summary of the VVP technique. Clarity of

this over arching objective would be greatly benefitted if both the 'normal' and VVP techniques were summarized in Section 2 and contrasted with the new regularization method (obviously with the in-depth details of regularization presented in Section 3). That is, what are the differences between the fitting methods and what will the new regularization method add.

Reply: The different algorithms are now 'named' and we expanded the discussion on the VVP. We introduced the 'all-sky' fit or standard analysis, the volume velocity processing as well as a 'packed' and 'full' wind retrieval, which differ in the way the weights are estimated and whether we employ a mesoscale regularization.

Comment: P.3, lines 13-16: What is a typical number of meteor trails which gives statistical uncertainty of 1-6 m/s at altitudes between 82 and 95 km?

Reply: We expanded this part of the paragraph and added some numbers. However, this numbers should not be taken as absolute measure as the obtained statistical uncertainties of the obtained winds are also reflect some other geophysical processes.

Comment: P.4, l.21: This is the first time azimuth and zenith are mentioned. The azimuth and zenith (elevation) angles, along with angle of arrival should be defined earlier with respect to the defined axes systems of: a) the radar(s)/links, and b) the local co-ordinates.

Reply: We added the convention used in this paper, after we introduced the angle of arrival.

Comment: P.5, l.20: How are values for p and ngamma_x determined for Equ. 1?

Reply: We added our reasoning behind those numbers in the paragraph. The value of p and also ngamma was estimated to ensure that a meteor at the edge of a grid cell enters the retrieval with a non-negligible weight. As the distance is measured in meteor a value of p=1 would give a meteor at 30 km distance already almost no weight.

Comment: P.6, l.22, Equ. 6: The terms defining the spatial derivative, time derivative?,

etc. in the smoothness matrix L are unclear. Please define and clarify.

Reply: We expanded the description of the L matrix and inserted a scheme outlining what we want to achieve. The matrix L is constructed based on the scheme shown for each wind component. The shown L –matrix here in just shows the spatial component. The temporal weight is inserted as weight in the sigma matrix.

Comment: P7, l.16: For the variance nsigma_iËĘ2, why not just use the measurement error?

Reply: We now describe in more detail what are the differences between both approaches and compare both retrievals.

Comment: P8, l.3: What do the authors mean by "optimal solution" for the regularization parameter nalpha? And how was the regularization parameter nalpha value obtained/ justified by being "estimated through several iterations"? Also, this optimal value of 0.014 is notthe global value typically used (which is nalpha = 0.1). Why is this?

Reply: Thanks for pointing at this inconsistency ion the first version. As we have two different retrieval algorithms called 'packed' and 'full' wind retrieval we looked for different strategies to find an optimal alpha. However, as it turned out with the 'full' retrieval that alpha=0.1 seems to be more robust, whereas for the packed retrieval sometimes also 0.01-0.02 provided reasonable solutions. The value of 0.1 seems to be more on the save side with respect to outliers or erratic measurements. The corresponding paragraph was updated to avoid this inconsistency.

Comment: P8, lines 3-5, Fig. 4: I like how the authors used extreme values of the regularization parameter nalpha to show different fits, but I have some concern on how different the fits appear. Although this may mostly be due to the winds in the plot on the right, with nalpha = 10ËĘ{-6}, not being scaled between the images. Why is this? And can they be re-scaled. Explain and clarify. Also, for the nalpha = 10ËĘ{-6} case, is this weak regularization essentially the 'normal' or 'standard' fitting method as nalpha

goes to zero? Once again, this relates to comparing the regularization technique to the other two fittings techniques ('normal' and VVP).

Reply: The alpha values shown here were really just picked to show extremes. In the case of the horizontally resolved wind solution there is no 'normal' fitting or standard solution in a least squares sense. The value of alpha= 10Ë̈Ę{-6} more or less presents the radial solution for each grid cell without a neighbor cell. We found not yet a solution to scale the arrows in a more consistent way. As the reviewer already pointed out, part of that is the arrow scaling.

Comment: P8, l.10, supplementary movies: Should not the regularization parameter nalpha be the same for all fits? Please justify and explain selecting different values of the regularization parameter.

Reply: The movies are redone. Somehow images from different runs using different alphas or estimated alphas were mixed. We are sorry for that mistake.

Comment: P9, l.10, Fig. 8: One would assume that the "all-sky fit as described above" and then presented in Fig. 8 would be the new regularization technique, but according to the Fig. 8 caption it is the 'standard' mean wind analysis. If this is the case, why is the new regularization technique not used?

Reply: We updated the figures and labeled the axis according to the introduced names.

Comment: Then on P9, l.28 the "all-sky fit" clearly refers to the 'normal' or 'standard' fit. Again, please clarify and standardize terminology to different fitting methods.

Reply: This is now clarified in the manuscript.

Comment: P.9, l.27 to P.10, l.7: This text validates that the mean 2D wind fit (regularization fit I assume) agrees well with previous accepted fitting techniques (all-sky or 'standard' and VVP fits). If the new regularization is the same as the accepted fits, what has been gained by this new technique. This should be related back to Section 2 and the benefits of using regularization should be elaborated.

Reply: This comparison should just underline that the retrieval does not add or change the mean wind estimates, which are also using many of the implemented techniques of the 2D wind retrieval. As suggested by the reviewer we added some more benefits of the new retrieval in the discussion and conclusion to underline the benefits of the new analysis.

Comment: P.10 Discussion Section, P.11, lines 25-29: Are there any other benefits to the small scale structures that are detectable using the new regularization scheme besides the behavior of the GW spectral slope? If so, list a few.

Reply: We expanded the list of potential benefits of the new technique. In particular, the synergy to other observations dealing with smaller scale structures is now also pointed out.

Technical corrections: Comment: P.5, l.27, Equ. 2: For 2nd term on RHS of equation it should be nsin(nphi_i) ËĘËĘËĘËĘ P.7, l.25: Do you mean "focus on horizontal winds", not "vertical" winds?

Reply: The mistake is corrected.

Please also note the supplement to this comment:
https://www.atmos-meas-tech-discuss.net/amt-2018-93/amt-2018-93-AC2-supplement.pdf

———————————————————

[Figure]

**Supplement:**

[revised manuscript text omitted]

$$

---

## Author Response (AR2)

Reply to Reviewer:

The authors thank the reviewer for helping to improve the submitted manuscript and finding all the typos.

**Comment 1:**

The defining of the different algorithms ('naming') presented in the manuscript is now much clearer. For even better clarity I strongly suggest the authors continue this in Sections 4, 5, 6, and 7. For example (version 3, not the difference version):

Comment:

P.12, l.12: 'A crucial part of the wind field analysis...', name of which one presented?

P.13, l.9: 'The introduced wind retrieval algorithm for...', which one?

....

and elsewhere in Sections 4, 5, 6, and 7.

**Reply:**

We updated the sections using the introduced namings. When no specific name is mentioned, the statement or sentence is valid for all the different retrieval versions.

**Comment 2:**

Equ. 7 (Equ. 2 in ver2) on P.6, l.21 (ver3): I believe you missed this in your revision.

The equation should be (see 2nd term in product of 2nd term of sum on RHS):

$v_{r,i} = u_j \cos(\phi_i) \sin(\theta_i) + v_j \sin(\phi_i) \sin(\theta_i) + ...$

^^^

NOT

$v_{r,i} = u_j \cos(\phi_i) \sin(\theta_i) + v_j \cos(\phi_i) \sin(\theta_i) + ...$

^^^

**Relpy:**

Done.